# Systemic and Local Medical or Surgical Therapies for Ear, Nose and/or Throat Manifestations in ANCA-Associated Vasculitis: A Systematic Literature Review

**DOI:** 10.3390/jcm12093173

**Published:** 2023-04-28

**Authors:** Roline M. Krol, Hilde H. F. Remmelts, Ruth Klaasen, Annelies Frima, E. Christiaan Hagen, Digna M. A. Kamalski, Marloes W. Heijstek, Julia Spierings

**Affiliations:** 1Department of Rheumatology & Clinical Immunology, University Medical Center Utrecht, 3584 CX Utrecht, The Netherlands; 2Department of Nephrology, Meander Medical Center, 3813 TZ Amersfoort, The Netherlands; 3Department of Rheumatology, Meander Medical Center, 3813 TZ Amersfoort, The Netherlands; 4Department of Otorhinolaryngology, Meander Medical Center, 3813 TZ Amersfoort, The Netherlands; 5Department of Otorhinolaryngology–Head and Neck Surgery, University Medical Center Utrecht, 3584 CX Utrecht, The Netherlands

**Keywords:** ANCA-associated vasculitis, GPA, EGPA, MPA, biologicals, csDMARDs

## Abstract

Background: Ear, nose and throat (ENT) manifestations are common in patients with antineutrophil cytoplasmic antibody (ANCA)-associated vasculitis (AAV), yet how to treat these manifestations remains controversial. Therefore, we systematically reviewed the literature on the efficacy of therapies on ENT manifestations in AAV. Methods: A systematic review was conducted in accordance with the PRISMA guidelines, searching Medline, Embase and Cochrane libraries, including clinical studies between January 2005 and January 2022, in adults with AAV and ENT involvement, reporting on the effects of local and systemic therapy. The critical appraisal was performed using tools provided by the Cochrane Library and the level of evidence (LoE) was scored according to the Oxford Centre for Evidence-based Medicine. Results: After screening 5609 identified studies, 136 full-text articles were assessed. Finally, 31 articles were included for critical appraisal and data-extraction. Nearly all studies (n = 29) were retrospective and scored low on LoE. The included studies evaluated local interventions (n = 11), glucocorticoids combined with conventional synthetic disease modifying anti-rheumatic drugs (csDMARDs) (n = 8), rituximab (n = 6), or mepolizumab (n = 6). Due to heterogeneity across studies meta-analysis was not performed. Four studies on mepolizumab for sinonasal symptoms (n = 92) showed response in 33–100% and relapse in 35%. Local therapy for subglottic stenosis was effective in 80–100% of patients in 11 studies (n = 157), but relapses were common (up to 83%). In five studies, hearing improvement was observed in 56–100%, with better outcomes when glucocorticoids were combined with csDMARDs compared to glucocorticoids only. Conclusion: Response rates of ENT manifestations varied widely in studies and relapses were observed frequently. Heterogeneity among studies impaired comparison.

## 1. Introduction

Anti-neutrophil cytoplasmic antibody (ANCA)-associated vasculitis (AAV) is a systemic autoimmune disease characterized by inflammation of the small- and medium-sized blood vessels [1,2]. In this heterogenous disease, organ system involvement varies among the different AAV subtypes: granulomatosis with polyangiitis (GPA), microscopic polyangiitis (MPA) and eosinophilic GPA (EGPA). Major organs, including kidneys and lungs, can be affected, as well as minor organs including ear, nose and throat (ENT) involvement [3]. ENT manifestations are reported in a majority of patients with AAV [3,4,5,6]. Nasal symptoms are present in 21.6–52.2%, sinus involvement in 30.4–33.8% and hearing loss and otitis in 10.7–18.5% [3,7]. Subglottic stenosis is present in approximately 1% of AAV patients [7]. Patients with GPA most often report ENT symptoms (72.3%). ENT disease has a negative impact on quality of life and can lead to permanent damage [8].

The guidelines advise treating patients with non-organ threatening disease, such as ENT symptoms, with methotrexate or mycophenolate mofetil in combination with glucocorticoids [9]. Furthermore, in the case of S. aureus carriage, treatment with trimethoprim/sulfamethoxazole could be considered [10]. These therapies, however, may not always resolve ENT symptoms sufficiently and up to 47% of patients experience ENT relapses [11,12,13,14,15,16,17,18,19,20,21,22,23,24,25]. Unfortunately, studies on systemic therapy mostly focus on outcomes for major organ involvement and often do not report results for ENT involvement specifically. As a result, the guidelines make no recommendations for the systemic treatment of ENT involvement in particular and there is little information available on management of hearing loss and subglottic stenosis. 

Therefore, this study aimed to systematically review evidence for the effect of systemic and local or surgical treatments on ENT symptoms in adult patients with AAV and ENT involvement.

## 2. Materials and Methods

### 2.1. Search Strategy and Selection Criteria

For this systematic literature review, a research question was formulated regarding the systemic and local treatment of ENT symptoms. The PICO-method (Population, Intervention, Comparison, Outcome) was used with AAV patients, with ENT involvement as population; local, surgical and systemic therapies as interventions; and ENT activity (defined as disease activity, relapse and damage) as outcomes (Appendix A).

A search string was designed including synonyms for the population and outcome (Appendix A). Synonyms for interventions were not included in the search string in order to yield as many potentially relevant records as possible. In order to be extensive in our systematic literature review without the risk of including outdated literature, we set the earliest date of literature to be included at 2005. The databases Medline (via Pubmed) and Embase were searched for articles published between January 2005 and January 2022 using this search string. Additionally, in Cochrane Library a search was performed with the terms “AAV, EGPA, Churg-Strauss, GPA, Wegener and MPA”, using the same time frame.

Inclusion criteria were studies evaluating therapies in patients with AAV and ENT involvement with a minimum age of 18 years. Exclusion criteria were studies that did not assess any therapies, animal studies, articles written in a language other than English, articles with no full-text available, congress abstracts, letters to editors, guidelines and case reports with less than five cases. The references of relevant articles were screened. Relevant new articles not retrieved in the search could be added by the committee. 

An initial screening of titles was undertaken by one researcher (RK), followed by a screening of abstracts from the remaining studies (BK, RK, HR, MH, JS). After this screening, all remaining articles were screened in full-text form. Both the screening of abstracts and the full-text screening were performed by two members of the committee independently. Disagreements between members of the committee were discussed. 

### 2.2. Interventions and Outcomes

Studies that assessed the effect of systemic immunosuppressive therapies, local therapies and surgical interventions were included (Appendix A). Outcomes reflecting treatment effects were: ENT disease activity (preferably described according to the Birmingham vasculitis activity score version 3 (BVAS-3)), relapse of ENT symptoms or damage (preferably described according to the vasculitis damage index (VDI)) [26].

### 2.3. Data-Extraction and Critical Appraisal

Data-extraction for all included articles was performed by two authors independently. Retrieved information from the articles included the name of the first author, publication year, country where the study was performed, number of patients included in the study, AAV type of the studied population, the intervention that was studied, other systemic therapies that were used simultaneously and outcome measures (disease activity, relapse and damage). 

The critical appraisal was performed using tools provided by the Cochrane Library, rating all included studies on validity [27]. All articles were scored with a level of evidence (LoE) according to the Oxford Centre for Evidence-based Medicine (Appendix A) [28]. All included articles were assessed independently by two members of the committee; discrepancies were resolved through discussion. 

This review was conducted in accordance with the Preferred Reporting Items for Systematic Review and Meta-Analysis (PRISMA) guidelines [29]. The study was registered with PROSPERO (CRD42020184663). There was no funding source for this study.

## 3. Results

A total of 5609 records were retrieved from the search. During the first screening, 484 duplicates were removed as well as 164 congress abstracts, 454 case reports and 4126 records not reporting on ENT in patients with AAV (Figure 1). After this screening, 371 articles were screened on title and abstract; 235 records were excluded. The remaining 134 records were assessed in full-text and a final 31 studies were included. An overview of all articles assessed in full-text can be found in Appendix A. The included studies were grouped based on ENT manifestations. The baseline characteristics of the included studies are shown in Table 1, Table 2, Table 3 and Table 4. 

### 3.1. Treatment of Sinonasal Manifestations

Seven studies (n = 406, AAV ENT patients with intervention n = 156) investigated the response of systemic therapies on sinonasal symptoms (Table 1). Except for one study by Holle et al., all studies included EGPA patients only. The level of evidence was 4 for all studies except the multicenter double-blind phase 3 trial (n = 136) by Wechsler et al. (LoE 1b). In this study, patients with relapsing or refractory EGPA were treated with subcutaneous mepolizumab 300 mg or placebo every four weeks in combination with standard care (glucocorticoids with or without conventional synthetic disease modifying antirheumatic drugs (csDMARDs)) for a duration of 52 weeks [14]. Sinonasal relapse was seen in 35% (n = 24) of the mepolizumab group compared to 51% (n = 35) of the placebo group, during a follow-up of 60 weeks. Of all the included patients with a high absolute eosinophilic count, remission was achieved for ≥24 weeks in 33% of patients treated with mepolizumab versus 0% in the placebo group (OR 26.10; 95% CI, 7.02–97.02). The efficacy of mepolizumab was lower in patients with a lower absolute eosinophilic count (21% vs. 7%, OR 0.95; 95% CI, 0.28–3.24). The study by Detoraki et al. prospectively followed eight patients treated with mepolizumab 100 mg every four weeks in combination with glucocorticoids for 12 months [30]. A significant decrease in sinonasal symptoms was reported. The mean sinonasal outcome test (SNOT-22) score decreased from 49 to 22 after 12 months and the mean total endoscopic polyp score (TENPS) decreased from 3.4 to 0.8 after 12 months. In the retrospective study by Rios-Garces et al., eleven patients with ENT involvement were treated with mepolizumab in combination with glucocorticoids and in some patients csDMARDs. Response to therapy was seen in 50% (n = 4) of patients with nasal polyps, in 33% (n = 1) of rhinitis patients and in 33% (n = 1) of patients with paranasal sinus involvement [31]. Much higher response rates were seen in another retrospective study including nine patients with sinonasal involvement in which response was seen in 100% of patients; however, the follow-up period was not reported [32].

A retrospective cohort study in 44 EGPA patients with chronic rhinosinusitis treated with csDMARDs and glucocorticoids showed remission in 21% (n = 9) and partial response in 32% (n = 14) during a mean follow-up of 4.54 years [19]. A much higher response rate was observed in a retrospective study with 17 EGPA patients with nasal polyposis treated with oral glucocorticoids (1 mg/kg) with or without csDMARDs and intranasal glucocorticoids for a duration of 12 weeks [12]. Remission, defined as the resolution of symptoms for at least six months, was reported in 82.3% (n = 14). All patients reported improvement of symptoms. In the retrospective study by Holle at al., 59 patients were treated with rituximab (RTX) for refractory GPA, including three patients with sinusitis [18]. Two patients (67%) showed response to treatment during a median follow-up of seven months. 

### 3.2. Treatment of Subglottic Manifestations

Twelve studies (n = 556, AAV ENT patients with intervention n = 165) reported on the effect of therapies for subglottic stenosis (SGS) in GPA patients, while the LoE was 4 in all studies (Table 2). 

Only one retrospective study investigated the effect of systemic therapies in patients with SGS. This study reported on the effect of RTX in 59 refractory GPA patients, including eight with SGS [18]. Patients were treated with four intravenous doses of 375 mg/m^2^ RTX in combination with 100 mg prednisolone with intervals of a week and followed for seven months. Patients were treated with one cycle of four RTX doses except for two patients who received two and three cycles. Any other immunosuppressive therapies could be continued. Complete remission, defined as absence of disease activity, was achieved in three (37.5%) of the eight patients with SGS, whereas 50% (n = 4) of patients had a >50% reduction in disease activity and the absence of new symptoms. 

The other eleven uncontrolled descriptive studies reported on combinations of local interventions including dilatation, intralesional glucocorticoids and surgical procedures [33,34,35,36,37,38,39,40,41]. Response to treatment was observed in 80–100% of patients and the mean number of procedures required was up to 3.5. Relapses were found in 38.5–83.3% of patients during a follow-up period ranging from two weeks to 20 years (Table 2). One study reported improvement of quality of life in 85% (n = 11) after surgery [42].

The study by Chen et al. studied differences in dilatation intervals for patients treated with different systemic immunosuppressive therapies compared to the patients not treated with that therapy [43]. Median dilatation interval in leflunomide- (n = 4) versus non-leflunomide-treated patients was 484 versus 155 days (*p* = 0.033). For rituximab, methotrexate and azathioprine, no significant differences were found. There was no correction for other therapies used previously or simultaneously. 

### 3.3. Treatment of Otitis and Inner Ear Dysfunction

Five studies (n = 336, AAV ENT patients with intervention n = 153) reporting on the effect of systemic immunosuppressants on otologic involvement were included. All studies were retrospective cohort studies and had a LoE of 4 except for the study by Okada et al. which had a LoE of 2b (Table 3). 

Three studies included patients with otitis media with AAV (OMAAV) [16,17,44]. OMAAV was defined as intractable otitis media with progressive hearing loss in AAV patients after the exclusion of other causes [24]. The fourth study was a case series of eleven patients with GPA and otologic symptoms including hearing loss (n = 10, 91%, conductive n = 3, sensorineural n = 2, mixed n = 5), otitis media with effusion (n = 10, 91%) and/or Eustachian tube dysfunction (n = 6, 55%). Patients were treated with a combination of methotrexate (MTX), anti-tumor necrosis factor therapy and glucocorticoids. All patients experienced an improvement of symptoms; a definition of this outcome was not reported [13]. 

In the retrospective study by Okada et al., patients refractory to other immunosuppressive therapies were treated with RTX in combination with glucocorticoids 0.5–1.0 mg/kg [16]. All six patients had a response to treatment, with a mean air conduction hearing gain of 22 dB and a bone conduction hearing gain of 11 dB. Harabuchi et al. studied OMAAV patients with otologic symptoms including hearing loss (n = 233, 99%), otorrhea (n = 120, 51%) otalgia (n = 93, 41%), tinnitus (n = 113, 51%), vertigo or dizziness (n = 74, 27%) and headache (n = 61, 26%). Patients treated with a combination of csDMARDs and glucocorticoids had a significantly better hearing improvement compared to patients treated with steroid monotherapy (68% vs. 56%, *p* < 0.01) [17]. Treatment with glucocorticoids in combination with csDMARDs was found to be an independent predictive factor for hearing improvement (OR 2.58, 95% CI 1.56–4.32, *p* = 0.0002) and lack of disease relapse (OR 1.90, 95% CI 1.07–3.42 *p* = 0.03). In two other studies evaluating patients (n = 19) on glucocorticoids and csDMARDs, improvement was seen in 81–100% [13,25]. The effect of mepolizumab was studied in one retrospective cohort in which six EGPA patients with eosinophilic otitis media showed a response to treatment in 83% (n = 5) [32].

One small study assessed patient-reported vestibular symptoms treated with glucocorticoids (n = 7) and glucocorticoids combined with intravenous cyclophosphamide 500 mg once a week (n = 3) for a non-specified period of time. Self-reported response to treatment was seen in 57.1% (n = 4) of the patients treated with glucocorticoids and in 100% (n = 3) of patients treated with glucocorticoids and cyclophosphamide [24]. 

### 3.4. Treatment of Non-Specified ENT Manifestations

An additional eight observational studies (n = 450, AAV ENT patients with intervention n = 293) reported on the response to therapy in patients with ENT symptoms. In these studies the specific ENT symptoms were not specified (Table 4), while the LoE was 2b-4. In four studies, 130 patients were treated with RTX. Three uncontrolled observational studies assessed the effect of RTX on ENT involvement in AAV patients with refractory or relapsing disease or with a contraindication to classic immunosuppressive therapies [20,21,22]. The study by Eriksson et al. prospectively followed nine AAV patients, including seven patients with ENT involvement [20]. Five patients were treated with four weekly infusions of 500 mg RTX (or 375 mg/m^2^ in one patient weighing 140 kg), two patients were treated with two weekly infusions of 500 mg RTX. All patients received prednisolone (daily dosage 5 to 40 mg per day) during treatment with RTX, and all but one patient received other immunosuppressives during RTX treatment, including mycophenolate, cyclophosphamide and azathioprine. All patients achieved complete or partial remission of AAV (86%, n = 6 and 14%, n = 1, respectively). During a follow-up period ranging from 6–25 months, two patients (28%) had an ENT relapse. In a retrospective study, 69 refractory EGPA patients were treated with RTX induction therapy followed by RTX maintenance therapy [22]. At each RTX infusion, patients were also treated with intravenous hydrocortisone 100 mg. During a follow-up of 24 months, 17.4% (n = 12) of the patients experienced ENT relapse. The third, retrospective, study included eleven patients with refractory GPA treated with four weekly RTX 375 mg/m^2^ combined with intravenous methylprednisolone. There was no significant decrease in ENT symptoms as scored in the BVAS but the authors did report a significant drop in daily glucocorticoid dose [21].

The study by Lally et al. retrospectively compared GPA patients with ENT involvement who received RTX (n = 51) with patients who did not (n = 48) [11]. Response to treatment was seen in 94.1% (n = 48) and there was absence of ENT activity during 92.4% of the observational period in RTX-treated patients compared to 53.7% in the non-RTX group (odds ratio 11.0, 95% confidence interval 5.5–22.0, *p* < 0.0001). Absence of ENT activity was seen in 58.9% for MTX-, 56.2% for cotrimoxazole- and 54.1% for azathioprine-treated patients.

Two retrospective studies reported on the effects of mepolizumab on ENT involvement. In the study by Bettiol et al., 138 patients with ENT involvement were treated with mepolizumab 100 mg or 300 mg every four weeks (n = 121 and n = 17, respectively) in combination with standard care (glucocorticoids in most patients and csDMARDs in some) (Table 4) [45]. In patients receiving 100 mg, ENT involvement decreased from 76.6% at baseline to 20.5% at 24 months (*p* < 0.001), for patients treated with 300 mg every four weeks, a decrease from 51.5% at baseline to 27.6% at 12 months was seen (*p* = 0.034). The second study reported six patients with ENT involvement treated with mepolizumab 300 mg every four weeks; in 50% (n = 3) of the patients, ENT manifestations were no longer present 12 months after mepolizumab treatment [46]. This study also reported on damage, using the vasculitis damage index (VDI). Before treatment with mepolizumab, chronic rhinosinusitis was present in six patients, after treatment with mepolizumab this increased to seven patients.

Another two studies retrospectively analyzed patients treated with glucocorticoids with or without csDMARDs. In one study, out of 28 GPA patients with ENT involvement treated with different csDMARDs and glucocorticoids, 95% (n = 20) achieved remission [15]. The case series by Yilmaz et al. reported remission in 100% (n = 15) of EGPA patients treated with glucocorticoids only [23]. The mean follow-up period was 1.7 years, during which none of the patients suffered an ENT relapse.
jcm-12-03173-t001_Table 1Table 1Overview of articles reporting on the treatment of sinonasal manifestations in AAV.First AuthorCountryPublication YearInterventionStudy DesignN of PatientsTotal/ENT Intervention/ENT ControlAAV TypeFollow-UpResultsLoE ^h^Validity ^h^ENT Disease ActivityENT RelapseSystemic therapyWechsler [14]International (9 countries)2017Intervention: MEPO + GC +/− csDMARDs ^a^Control: Placebo + GC +/− csDMARD ^a^randomized, placebo-controlled,double-blind, parallel-group, phase 3 trialTotal: 151Intervention: 64Control: 64EGPAAll included pt 60 wn/rRelapseIntervention: 35% (n = 24)Control: 51%1b+Rios-Garces [31]Spain2021Intervention: MEPO + GC +/− csDMARDs ^b^Control: noneRetrospective cohortTotal: 56Intervention: 11Control: noneEGPAMedian 3.19 y(3 m–5.6 y)Response: Nasal polyps 50% (n = 4), Rhinitis 33% (n = 1),Paranasal sinus involvement 33% (n = 1)n/r4+Tsurikisawa [32]Japan2021Intervention: MEPO + GC +/− csDMARDs ^c^Control: noneRetrospective cohortTotal: 59Intervention: 9Control: noneEGPAn/rResponse in 100% of patients (n = 9)n/r4−Detoraki [30]Italy2021Intervention: MEPO + GC ^d^Control: noneProspective cohortTotal: 8Intervention: 8Control: noneEGPAAll included pt 12 mDecrease mean SNOT-22 49 (t = 0) to 22 (t = 12 m), decrease in mean TENPS 3.4 (t = 0) to 0.8 (t = 12 m)n/r4−Holle [18]Germany2012Intervention: RTX + GC +/− CYC ^e^Control: noneRetrospective cohortTotal: 59Intervention: 3Control: noneGPAMedian 7 m(4–58 m)Response 67% (n = 2)n/r4+/−Low [19]United States2020Intervention: csDMARDs + GC^f^ Control: noneRetrospective cohortTotal: 44Intervention: 44Control: noneEGPAMean 4.54 y (SD 4.98)Remission 21% (n = 9), response 32% (n = 14), no response 21% (n = 9)n/r4+/−Bacciu [12]Italy2008Intervention: GC + intranasal GC +/− csDMARDs ^g^Control: noneRetrospective case seriesTotal: 29Intervention: 17Control: noneEGPAMean 43 m(12 m–74 m)Remission 82.3% (n = 14), improvement of symptoms 100% (n = 17)n/r4+AAV: ANCA-associated vasculitis, AZA: azathioprine, csDMARDs: conventional synthetic disease modifying antirheumatic drugs, CYC: cyclophosphamide, EGPA: eosinophilic granulomatosis with polyangiitis, ENT: ear, nose and throat, GC: glucocorticoids, GPA: granulomatosis with polyangiitis, LoE: level of evidence, m: months, MEPO: mepolizumab, MTX: methotrexate, N: number, n/r: not reported, pt: patients, RTX: rituximab, SD: standard deviation, SNOT: Sino-Nasal Outcome Test, TENPS: Total Endoscopic Polyp Score, w: weeks, y: years. ^a^ GC dose was not reported. csDMARDs in intervention group n = 41, in control n = 31, not defined which therapies were used. ^b^ GC dose prednisone 1 mg/kg/day, intravenous CYC n = 4 (500–1000 mg per infusion, 8–12 infusions per patient), MTX n = 2, AZA n = 2. ^c^ GC dose mean prednisolone dose 12.7 mg/day (not reported if all patients received GC), mepolizumab dose was not reported, AZA n = 5, cyclosporine n = 3, MTX n = 12, RTX n = 2. ^d^ GC dose was not reported. ^e^ GC dose prednisolone 1 mg/kg/day and 100 mg prednisolone at every RTX infusion. No information on N of patients treated with CYC during RTX treatment, CYC treatment: oral dose 2 mg/kg/day, intravenous 3 doses 15–20 mg/kg at weekly intervals. ^f^ GC dose was not reported. AZA n = 12, leukotriene receptor antagonist n = 13, CYC n = 12, MTX n = 6, biological n = 2. ^g^ GC dose prednisone 1 mg/kg/day, CYC n = 8, MTX n = 1, AZA n = 1. ^h^ Validity was scored using tools provided by the Cochrane library [27]. Level of evidence was scored according to the Oxford Centre for Evidence-based Medicine [28].
jcm-12-03173-t002_Table 2Table 2Overview of articles reporting on the treatment of subglottic manifestations in AAV.First AuthorCountryPublication YearInterventionStudy DesignN of PatientsTotal/ENT Intervention/ENT ControlAAV TypeFollow-UpResultsLoE ^h^Validity ^h^ENT Disease ActivityENT RelapseSystemic therapyHolle [18]Germany2012Intervention: RTX + GC +/− CYC ^a^Control: noneRetrospective cohortTotal: 59Intervention: 8Control: noneGPAMedian 7 m (4–58 m)complete remission 37.5% (n = 3) response 50% (n = 4)refractory 12.5% (n = 1)n/r4+/−**Local interventions (in combination with systemic therapy)**Zammit [33]United Kingdom2021Intervention: dilatation + GC + i.v. CYC or RTX ^b^Control: noneRetrospective cohortTotal: 20Intervention:20Control: noneGPAMean 61.2 m (15.7–201.5 m)Remission 90% (n = 18)Relapse 10% (n = 2)4+/−Schokkenbroek [34]Netherlands2008Intervention: dilatation ^c^Control: noneRetrospective cohortTotal:25Intervention:9Control: noneGPAMean 25.4 m +/− 41.1 mn/r77.8% (n = 7)4−Taylor [35]United States2013Intervention: dilatation +/− local/intralesional GC +/− csDMARD ^d^Control: noneRetrospective cohortTotal: 39Intervention: 15Control: noneGPAMean 8.2 y, median 9.9 yn/rMean n of procedures/pt 3.534−Wolter [36]Canada2010Intervention: Dilatation + intralesional GC ^c^Control: noneRetrospective cohortTotal: 12Intervention: 8Control: noneGPAn/rn/rMean n of procedures/pt 3.37, mean symptom control 11.9 months4−Fijolek [37]Poland2016Intervention: Dilatation + intralesional GC +/− systemic GC and csDMARD ^e^Control: noneRetrospective cohortTotal: 250Intervention: 34Control: noneGPAMedian 7 y (2 w–20 y)88.2% (n = 30) response to treatmentMedian n of procedures/pt 1,median response interval 34 monthsRelapse in pt with systemic treatment in 32% (n = 11)4+/−Nouraei [38]United Kingdom2008Intervention ± Dilatation + intralesional GC + laser surgery ^c^Control: noneRetrospective cohortTotal: 18Intervention: 18Control: noneGPA5–38 mn/rMedian n of procedures/pt 1, mean intervention-free interval 26.1 months4+/−Carnevale [39]Spain2019Intervention: Dilatation + laser surgery ^c^Control: noneRetrospective case seriesTotal: 19Intervention: 5Control: noneGPAn/r80.0% (n = 4) response to treatmentn/r4−Costantino [40]United States2018Intervention: laryngotracheal resection + reconstruction +/− GC +/− csDMARD/biological ^f^Control: noneRetrospective case seriesTotal:11Intervention: 11Control: noneGPAMedian 10.9 y (4 m–28 y)91% (n = 10) response to treatment55% (n = 6) required additional local treatment4+/−Arebro [42]Sweden2012Intervention: micro larynx surgery ^c^Control: noneRetrospective case seriesTotal: 13Intervention: 13Control: noneGPAMean 3.5 y, 1.5 y–6.5 y100% (n = 13) response to treatment, 85% (n = 11) higher QoL38.5% (n = 5) relapsed4−Solans-Laque [41]Spain2008Intervention: Surgery or dilatation + intralesional GC +/− systemic GC +/− csDMARD ^g^Control: noneRetrospective case seriesTotal: 51Intervention: 6Control: noneGPAMean 71.3 m, 12 m–180 mn/r83.3% (n = 5) relapsed4−Chen [43]United States2020Intervention: dilatation + biological or csDMARDControl: dilatation + different csDMARDsRetrospective cohortTotal: 39Intervention: 18Control: 21GPAn/rMedian dilatation intervalRTX maintenance (n = 3) 153 d vs 80 d in non-RTX, MTX (n = 7) 259 d vs. 174 d in non-MTX, AZA (n = 4) 177 d vs. 394 d in non-AZA, LEF (n = 4) 484 d vs. 155 d in non-LEFn/r4−AAV: ANCA-associated vasculitis, AZA: azathioprine, csDMARDs: conventional synthetic disease modifying antirheumatic drugs, CYC: cyclophosphamide, ENT: ear, nose and throat, GC: glucocorticoids, GPA: granulomatosis with polyangiitis, LEF: leflunomide, LoE: level of evidence, m: months, MTX: methotrexate, N: number, n/r: not reported, pt: patients, QoL: quality of life, RTX: rituximab, SGS: subglottic stenosis, w: weeks, y: years, ^a^ GC dose prednisolone 1 mg/kg/day and 100 mg prednisolone at every RTX infusion. No information on N of patients treated with CYC during RTX treatment, CYC treatment: oral dose 2 mg/kg/day, intravenous 3 doses 15–20 mg/kg at weekly intervals. ^b^ GC dose was not reported. i.v. CYC n = 11 (3 doses every 2 weeks, followed by 7 doses every 3 weeks), RTX n = 10 (2 weekly doses of 1000 mg). ^c^ use of systemic therapies was not described. ^d^ local GC in 31/48 dilatations, systemic GC n = 13 (GC dose was not reported), MTX n = 11, CYC n = 9. ^e^ GC dose not reported. GC + CYC or MTX n = 21. ^f^ GC n = 2, MTX n = 1, RTX n = 3, AZA n = 2, adalimumab n = 1, tacrolimus n = 1. ^g^ GC n = 4 (GC dose was not reported), CYC n = 4 (dose was not reported, duration of therapy 18–24 months), AZA n = 2, mycophenolate mofetil n = 1. ^h^ Validity was scored using tools provided by the Cochrane library [27]. Level of evidence was scored according to the Oxford Centre for Evidence-based Medicine [28].
jcm-12-03173-t003_Table 3Table 3Overview of articles reporting on the treatment of otitis and inner ear dysfunction in AAV.First AuthorCountryPublication YearSystemic TherapyStudy DesignN of PatientsTotal/ENT Intervention/ENT ControlAAV TypeFollow- UpResultsLoE ^g^Validity ^g^ENT Disease ActivityENT RelapseOtitis media/Hearing lossOkada [16]Japan2019Intervention: RTX + GC ^a^Control: GC +/− i.v. CYC, AZA ^a^Retrospective cohortTotal: n = 23Intervention: n = 6Control: n = 17AAVn/rResponse, mean hearing gain AC/BCIntervention: 100%, 22 dB /11 dBControl: 100%, 21 dB / 10 dBn/r2b+/−Harabuchi [17] Japan2017Intervention: csDMARDs + GC ^b^Control: GC ^b^Retrospective cohortTotal: n = 235Intervention: n = 122Control: n = 101AAVMedian 24 m(Q25–75: 11 m–72 m)Hearing improvement rate Intervention: 68%Control: 56%RelapseIntervention: 36% (n = 45)Control: 47% (n = 47)4−Yoshida [25]Japan2014Intervention: CYC + GC ^c^Control: noneRetrospective case seriesTotal: n= 8Intervention n = 8Control: noneAAV12 m–96 mHearing improvementIntervention: 81% (n = 16) earsRelapseIntervention: 0% (n = 0)4+/−Sahyouni [13] United States2019Intervention: MTX + GC + aTNF ^d^Control: noneRetrospective case seriesTotal: n= 11Intervention: n = 11Control: noneGPAn/rimprovement of otologic symptomsIntervention: 100% (n = 11)n/r4−Tsurikisawa [32] Japan2021Intervention: MEPO + GC +/− csDMARDs ^e^Control: noneRetrospective cohortTotal: n = 59Intervention: n = 6Control: noneEGPAn/rResponse in 83% (n = 5)n/r4−**Vestibular symptoms**Morita [24] Japan2017Intervention: i.v. CYC + GC ^f^Control: GC ^f^Retrospective cohortTotal: n = 31Intervention: n = 3Control: n = 7AAVMedian 26 m(1 m–127 m)ResponseIntervention: 100% (n = 3)Control: 57.1% (n = 4)n/r4+/−AAV: ANCA-associated vasculitis, aTNF: anti-tumor necrosis factor agents, AZA: azathioprine, csDMARDs: conventional synthetic disease modifying antirheumatic drugs, CYC: cyclophosphamide, dB: decibel, ENT: ear, nose and throat, GC: glucocorticoids, GPA: granulomatosis with polyangiitis, i.v.: intravenous, LoE: level of evidence, m: months, MEPO: mepolizumab, MTX: methotrexate, N: number, n/r: not reported, pt: patients, Q: quartile, RTX: rituximab, w: weeks, y: years. ^a^ GC intravenous methylprednisolone 3 days 1000 mg/day n = 16, prednisolone 0.5–1.0 mg/kg/day in all patients. CYC 500 mg intravenous every 2–4 weeks n = 9, AZA n = 12. ^b^ GC dose was not reported. CYC n = 97 (oral n = 69, intravenous n = 28), AZA n = 11, MTX n = 6, ciclosporin n = 3, Tacrolimus n = 3. ^c^ GC dose prednisolone 30–40 mg/day, methylprednisolone 1000 mg/day 3 days n = 1. CYC oral 50 mg/day n = 7, intravenous n = 2. ^d^ GC dose prednisone 1 mg/kg/day up to 80 mg/day, MTX 0.3 mg/kg/week. ^e^ GC dose mean prednisolone dose 12.7 mg/day (not reported if all patients received GC), mepolizumab dose was not reported, AZA n = 5, cyclosporine n = 3, MTX n = 12, RTX n = 2. ^f^ GC dose prednisolone 20–60 mg/day, i.v. CYC 500 mg/week. ^g^ Validity was scored using tools provided by the Cochrane library [27]. Level of evidence was scored according to the Oxford Centre for Evidence-based Medicine [28].
jcm-12-03173-t004_Table 4Table 4Overview of articles reporting on the treatment of non-specified ENT manifestations in AAV.First AuthorCountryPublication YearSystemic TherapyStudy DesignN of PatientsTotal/ENT Intervention/ENT ControlAAV TypeFollow-UpResultsLoE ^i^Validity ^i^ENT Disease ActivityENT RelapseLally [11]United States2014Intervention: RTX +/− GC ^a^Control: csDMARDs +/− GC ^a^Retrospective cohortTotal: 99Intervention: 51Control: 48GPAn/rAbsence of ENT activity during % of observational periodIntervention: 92.4%, Control: 53.7%More absence of ENT activity in intervention group OR 12.0, *p* < 0.001n/r2b+/−Eriksson [20]Sweden2005Intervention: RTX + GC +/− csDMARDs ^b^Control: noneRetrospective case seriesTotal: 9Intervention: 7Control: noneAAV6 m–25 mremission 86% (n = 6), partial remission 14% (n = 1), drop in daily GC dose28% (n = 2)4+Malm [21]United States2014Intervention: RTX + GC ^c^Control: noneRetrospective case seriesTotal: 11Intervention: 11Control: noneGPAMean 23.5 m (6 m–48 m)drop in daily GC dosen/r4+/−Teixeira [22]United Kingdom2019Intervention: RTX + GC ^d^Control: noneRetrospective cohortTotal: 69Intervention: 61Control: noneEGPAIn all pt24 mn/r17.4% (n = 12)4+Bettiol [45]International (8 countries)2021Intervention: MEPO 100 mg/4 w + standard care ^e^Control: MEPO 300 mg/4 w + standard care ^e^Retrospective cohortTotal: 203Intervention: 121Control: 17EGPA3 m–24 mIntervention: ENT involvement decreased from 76.6% to 20.5% at 24 mControl: ENT involvement decreased from 51.5% to 27.6% at 12 mIntervention: 15.8% (n = 25)Control: 12.2% (n = 4)2b+/−Ueno [46]Japan2021Intervention: MEPO 300 mg/4 w + standard care ^f^Control: noneRetrospective cohortTotal: 16Intervention: 6Control: noneEGPAIn all pt12 mResponse 50% (n = 3)n/r4+Knopf [15]Germany2015Intervention: GC +/− csDMARDs ^g^Control: noneRetrospective case seriesTotal: 28Intervention: 21Control: noneGPAMean 38 m (8 m–56 m)Remission 95% (n = 20)n/r4−Yilmaz [23]Turkey2017Intervention: GC +/− csDMARDs ^h^Control: noneRetrospective case seriesTotal: 15Intervention: 15Control: noneEGPAMean 1.7 y (0.5 y–2 y)Remission 100% (n = 15)n/r4−AAV: ANCA-associated vasculitis, AZA: azathioprine, csDMARDs: conventional synthetic disease modifying antirheumatic drugs, CYC: cyclophosphamide, EGPA: eosinophilic granulomatosis with polyangiitis, ENT: ear, nose and throat, GC: glucocorticoids, GPA: granulomatosis with polyangiitis, LoE: level of evidence, m: months, MMF: mycophenolate mofetil, MTX: methotrexate, N: number, n/r: not reported, pt: patients, RTX: rituximab, y: years. ^a^ MTX at 197 visits, CYC at 55 visits (n/r whether oral of intravenous), AZA at 98 visits. The number of patients receiving GC was not reported, mean prednisone dose was 7.7 mg/day in the RTX group and 5.9 mg/day in the control group. ^b^ GC daily prednisolone dose 5–40 mg/day. MMF n = 5, AZA n = 1, CYC n = 2 (n/r whether oral of intravenous). ^c^ GC dose was not reported. ^d^ GC median daily dose prednisolone 12.5 mg, 100 mg hydrocortisone at every RTX infusion. ^e^ immunosuppressive treatment was not described specifically for patients with ENT involvement, in the overall study population: GC 96% (n = 194) median prednisone dose 10 mg/day, MTX 19% (n = 38), AZA 11% (n = 23), MMF 9% (n = 18), ciclosporin 1% (n = 2), RTX 11% (n = 23), intravenous immunoglobulin 6% (n = 12), other immunosuppressants 3% (n = 5). ^f^ GC median prednisone dose 8 mg/day, AZA n = 6, MTX n = 5, tacrolimus n = 1. ^g^ MTX n = 8, AZA n = 4, CYC n = 12 (n/r whether oral of intravenous), MMF n = 5, RTX n = 5, leflunomide n = 1. GC dose was not reported. ^h^ MTX n = 1, GC methyl prednisolone 2–12 mg/day. ^i^ Validity was scored using tools provided by the Cochrane library [27]. Level of evidence was scored according to the Oxford Centre for Evidence-based Medicine [28].


## 4. Discussion

This systematic literature review evaluated literature on the effect of local interventions and systemic treatment on ENT involvement in patients with AAV. Results were presented per ENT manifestation to provide a practical overview for clinicians (a summary of findings can be found in Table 5).

ENT manifestations were treated with a variety of immunosuppressive regimens most of which included therapy with glucocorticoids in combination with RTX, cyclophosphamide, mepolizumab or a csDMARD. Response to treatment was high in most studies (57.1–100%) but relapses were observed frequently. The addition of a csDMARD improved response rates compared to treatment with glucocorticoids only. Studies comparing responses to csDMARDs versus RTX reported no differences, but due to heterogeneity across studies with regard to treatment and outcome measures, no meta-analysis could be performed.

A relatively large number of studies (n = 11) reported on local interventions of SGS. Concurrent use of systemic therapies was described in six studies. Local intervention was found to be effective in nearly all patients but again relapse rates were high and most patients required multiple procedures. Local intervention may result in a good but temporary response. Therefore, insight into the effect of systemic therapies or maintenance therapy to prevent the necessity of local interventions for SGS is needed. The efficacy of RTX on SGS was described in one small study (n = 8) only and reported a very modest result. The limited data we found on efficacy of different systemic therapies in SGS warrant further research on this matter.

Studies reporting on otitis and inner ear dysfunction reported similar outcomes in patients treated with glucocorticoids and RTX or csDMARDs. The two studies comparing treatment with glucocorticoids versus glucocorticoids in combination with csDMARDs showed less relapses, a higher hearing improvement rate and a higher response rate in patients treated with csDMARDs and glucocorticoids. Of note, studies investigating the effect of systemic therapies on otologic manifestations were predominantly conducted in Japan. In these studies, inclusion criteria of the OMAAV study group of the Japan otologic society were used [47]. No studies from Europe or the USA have used these criteria to define their study population. It is therefore difficult to compare western and non-western studies. The use of identical criteria would enable comparison of future studies.

In this systematic literature review, we found a limited number of studies with mostly small patient numbers. Overall, the level of evidence of the included studies was limited, and except for one double-blind phase 3 trial, all studies were case series or cohort studies, with mostly a retrospective design. Only six out of 31 studies scored high on validity. In multiple studies, a clear description of the definitions used for outcomes or ENT involvement was lacking. In ten studies, ENT activity was not the primary outcome measure; therefore, these studies were not powered to demonstrate an effect on ENT activity of the intervention studied. Only one study assessed quality of life. However, this study did not use any validated questionnaires, such as the AAV-PRO (ANCA-associated vasculitis patient reported outcomes) or the EuroQoL [48]. Furthermore, due to heterogeneity across the studies with regard to treatment and outcome measures, no meta-analysis could be performed and robust recommendations for optimal treatment cannot be made. Standardized definitions of ENT involvement would enable better comparison in the future.

Lastly, we defined damage as an outcome measure of interest. Unfortunately, only one of the included studies used this endpoint. As a result, no recommendations with regard to therapy to prevent damage can be made either.

Most studies included in this systematic review researched biologicals or cyclophosphamide, whereas the recently published guideline by the American College of Rheumatology advised against treating patients with non-severe disease (such as ENT involvement) with RTX or cyclophosphamide. For EGPA, mepolizumab was recommended over treatment with csDMARDs, RTX or cyclophosphamide. Except for the advice not to treat SGS with local therapy only, no recommendations specifically for the treatment of ENT involvement were made [49]. The limited evidence on the effect of systemic therapy we found in this review impaired us from making strong recommendations on how to treat different ENT symptoms. Furthermore, except for local therapies for SGS, no studies on the effect of local therapies for ENT involvement were found. There have been studies on the effect of trimethoprim/sulfamethoxazole on ENT symptoms in AAV patients [50]. However, these studies were performed during a time with different treatment guidelines and before rituximab was registered as a therapy for AAV. The limited number of studies on the effect of local therapies in addition to current systemic treatment and the high number of relapses indicate the need for further prospective controlled studies on the effect of both local and systemic therapies on ENT involvement in AAV. A multidisciplinary approach to ENT involvement in AAV is of great importance for both the optimal treatment of patients and for further research on this subject. In order to objectify ENT involvement as well as the effect of therapies on ENT involvement, an otorhinolaryngologist should be involved in the treatment of all AAV patients with ENT involvement.

## 5. Conclusions

In conclusion, in this review we systematically reviewed the evidence on management of ENT involvement in AAV patients. We found high response rates as well as frequent relapses in patients treated with csDMARDs, CYC, RTX or MEPO. Heterogeneity among the studies impaired comparison. Further, more controlled studies, specifically focusing on ENT involvement, are needed to better guide the management of ENT symptoms in AAV patients.

## Figures and Tables

**Figure 1 jcm-12-03173-f001:**
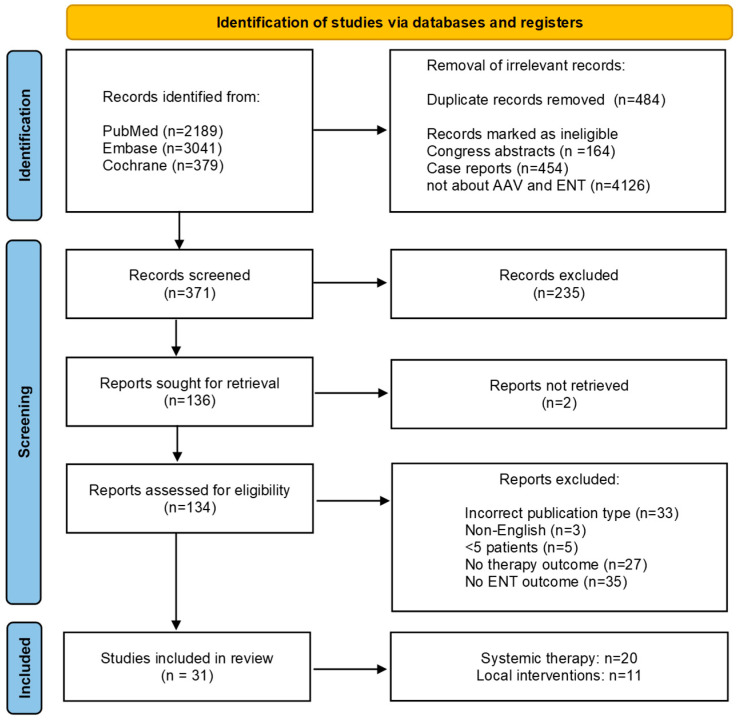
Flowchart of article selection.

**Table 5 jcm-12-03173-t005:** Summary of findings per manifestation.

Sinonasal manifestations	Local therapy	No information available
	Systemic therapy	Varying results in patients treated with GC in combination with a csDMARD (remission 21–82%) and patients treated with GC in combination with MEPO (relapse 35%, response in 33–100%)
Subglottic manifestations	Local therapy	Response of 80–100% in patients treated with dilatation therapy, intralesional GC, surgery or a combination of these therapies. Relapses were seen in 38–83% of patients with mean N of procedures per patient up to 3.5
	Systemic therapy	One study reporting complete remission in 38% of patients treated with GC and RTX
Otitis and inner ear dysfunction	Local therapy	No information available
	Systemic therapy	Hearing improvement in 68–100% of patients treated with csDMARDs combined with GC compared to 56–57% in patients treated with GC alone. Hearing gain in 100% of patients treated with GC in combination with either RTX or csDMARDs. Response in 83% of patients treated with MEPO in combination with GC with or without csDMARDs
ENT manifestations not specified	Local therapy	No information available
	Systemic therapy	Response in 86–100% of patients treated with csDMARDs and glucocorticoids, relapses were observed in 17–28%. Decrease in ENT involvement from 77% to 21% in patients treated with MEPO with or without GC and csDMARDs, relapse in 16%

csDMARDs: conventional synthetic disease modifying antirheumatic drugs, ENT: ear, nose and throat, GC: glucocorticoids, MEPO: mepolizumab, N: number, RTX: rituximab.

## Data Availability

No new data were created or analyzed in this study. Data sharing is not applicable to this article.

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
