# Peer review of "Systemic and Local Medical or Surgical Therapies for Ear, Nose and/or Throat Manifestations in ANCA-Associated Vasculitis: A Systematic Literature Review"

_jcm, 2023, doi:10.3390/jcm12093173_

Round 1
Reviewer 1 Report
The article is a very interesting systematic literature review about systemic and local medical or surgical therapies for ENT manifestations in ANCA-associated vasculitis. Unfortunately heterogeneity among studies impaired comparison.
Author Response
We would like to thank the reviewer for the positive feedback. It is indeed unfortunate that the large variation in interventions and outcome measures across studies limited comparison. We emphasized the importance of standardized definitions of ENT involvement in the discussion.
Reviewer 2 Report
I would like to congratulate the authors for this systematic review: well written, well-conducted from a methodological point of view and interesting and pleasant to read. It also highlights the lack of randomized controlled trials in ENT manifestations in ANCA associated vasculitides.
I have only some minor comments:
- Methods: Why did you put as a temporal limit of studies screening the year 2005?
- When possible, I would emphasize when studies are randomized controlled trials and when are observational. For example, in paragraph 2 (treatment of subglottic manifestations), you might outline the study design of the paper referenced [18], Holle et al.
- Just a curiosity: in tables 1-4 which were the criteria for which validity was scored with +, - or +/-? Only LoE? Or other factors were taken into account?
- In table 1, check carefully all the references. Low 2020 has the reference [18] but should be [19] according to the actual reference list. Right?
Thank you
Author Response
I would like to congratulate the authors for this systematic review: well written, well-conducted from a methodological point of view and interesting and pleasant to read. It also highlights the lack of randomized controlled trials in ENT manifestations in ANCA associated vasculitis.
We would like to thank the reviewer for the comments and positive feedback.
I have only some minor comments:
- Methods: Why did you put as a temporal limit of studies screening the year 2005?
Older literature may lose its relevance. Nevertheless older data can be informative when published data is scarce. Therefore, we included literature originating from the past 15 years at time we started our systemic literature review, that is since 2005.
We added an explanation of this selection in the method section of the paper.
Page 2, line 73: ´In order to be extensive in our systematic literature review without the risk of including outdated literature, we set the earliest date of literature to be included at 2005.´
- When possible, I would emphasize when studies are randomized controlled trials and when are observational. For example, in paragraph 2 (treatment of subglottic manifestations), you might outline the study design of the paper referenced [18], Holle et al.
We agree that in the results section the study design is not mentioned for all studies. We altered this in the following sentences:
- Paragraph 1, page 4, line 181 ‘A much higher….’
- Paragraph 2, page 4, line 194 ‘Only one retrospective ….’
- Paragraph 3, page 5, line 233 ‘In the retrospective ….’
- Paragraph 4, page 6, line 286 ‘Two retrospective studies ….’
- Just a curiosity: in tables 1-4 which were the criteria for which validity was scored with +, - or +/-? Only LoE? Or other factors were taken into account?
Other factors were taken into account, as mentioned in the methods paragraph 3. We performed the critical appraisal and scored all articles on validity using tools provided by the Cochrane Library. These tools are checklists that are based on the publication R. J. P. M. Scholten, M. Offringa en W. J. J. Assendelft (Red.), Inleiding in evidence-based medicine, https://doi.org/10.1007/978-90-368-1978-7_4
We added ‘Validity was scored using tools provided by the Cochrane library. [27] Level of evidence was scored according to the Oxford Centre for Evidence-based Medicine. [28]’ to Tables 1-4.
- In table 1, check carefully all the references. Low 2020 has the reference [18] but should be [19] according to the actual reference list. Right?
We agree there was an error in the references in Tables 1-4, we corrected the reference numbers.
Reviewer 3 Report
The authors have tackled interesting topics and often need to be more comprehensively presented in ENT. There are only a few minor recommendations.
1. Treatment of sinonasal manifestations P4: total n of patient n=405 may be shortened into n=406
2. Treatment of subglottis manifestations P4: same as above
3. P6 at the bottom: »This section may be divided …« may not be the text the authors intended to include?
4. P7 Table 1, Detoraki, Itlay, maybe Italy?
5. P11 Table 3: Okada and columns 1-7 are in bold, but there is no description of this meaning.
6. P13 Table 4 contnd: d immunosuppressive – probable double spacing
7. P14 Table 5: Sinonasal manifestations: Local therapy Response to surgery. There is no scarce data to support this in sinonasal manifestations.
8. P14 Table 5: otitis and inner ear dysfunction with a mild recommendation to change in Otitis …
9. P15 Paragraph 3: In the only study reporting … a mild recommendation to make the sentence more understandable to the reader.
Author Response
The authors have tackled interesting topics and often need to be more comprehensively presented in ENT. There are only a few minor recommendations.
We would like to thank the reviewer for the feedback and for the careful study of the manuscript.
- Treatment of sinonasal manifestations P4: total n of patient n=405 may be shortened into n=406
- Treatment of subglottis manifestations P4: same as above
We agree with the reviewer and altered this in all parageraphs in the result section:
- page 4, line 154
- page 4, line 191
- page 5, line 221
- page 5, line 255
- P6 at the bottom: »This section may be divided …« may not be the text the authors intended to include?
We would like to thank the reviewer for noticing. We removed the sentence.
- P7 Table 1, Detoraki, Itlay, maybe Italy?
We corrected the spelling error.
- P11 Table 3: Okada and columns 1-7 are in bold, but there is no description of this meaning.
This is not intentional, in the version we submitted this was not present. We will make sure the lay-out will be correct in the publicized version.
- P13 Table 4 contnd: dimmunosuppressive – probable double spacing
We corrected the double spacing.
- P14 Table 5: Sinonasal manifestations: Local therapy Response to surgery. There is no scarce data to support this in sinonasal manifestations.
We adjusted the text accordingly.
- P14 Table 5: otitis and inner ear dysfunction with a mild recommendation to change in Otitis …
We changed this accordingly.
- P15 Paragraph 3: In the only study reporting … a mild recommendation to make the sentence more understandable to the reader.
We altered the sentence to: ‘Only one study assessed quality of life. However, this study did not use any validated questionnaires, such as the AAV-PRO (ANCA-associated vasculitis patient reported outcomes) or the EuroQoL.’ (page 17, line 416)